# Red Blood Cell Exchange as a Valid Therapeutic Approach for Pregnancy Management in Sickle Cell Disease: Three Explicative Cases and Systematic Review of Literature

**DOI:** 10.3390/jcm12227123

**Published:** 2023-11-16

**Authors:** Caterina Giovanna Valentini, Claudio Pellegrino, Sara Ceglie, Vincenzo Arena, Francesca Di Landro, Patrizia Chiusolo, Luciana Teofili

**Affiliations:** 1Dipartimento di Diagnostica per Immagini, Radioterapia Oncologica ed Ematologia, Fondazione Policlinico Universitario “A. Gemelli” IRCCS, 00168 Rome, Italy; 2Sezione di Ematologia, Dipartimento di Scienze Radiologiche ed Ematologiche, Università Cattolica del Sacro Cuore, 00168 Rome, Italy; 3Sezione di Patologia, Dipartimento di Scienze della Salute della Donna, del Bambino e di Sanità Pubblica, Fondazione Policlinico Universitario A. Gemelli IRCCS, Istituto di Anatomia Patologica, Università Cattolica Del Sacro Cuore, 00168 Rome, Italy

**Keywords:** sickle cell disease, pregnancy, erythrocytapheresis, transfusion, HbS

## Abstract

Pregnancy in women with sickle cell disease (SCD) is a high-risk situation, especially during the third trimester of gestation and in the post-partum period, due to chronic hypoxia and vaso-occlusive phenomena occurring in the maternal–fetal microcirculation: as a result, unfavorable outcomes, such as intra-uterine growth restriction, prematurity or fetal loss are more frequent in SCD pregnancies. Therefore, there is a consensus on the need for a strict and multidisciplinary follow-up within specialized structures. Transfusion support remains the mainstay of treatment of SCD pregnancies, whereas more targeted modalities are still controversial: the benefit of prophylactic management, either by simple transfusions or by automated red blood cell exchange (aRBCX), is not unanimously recognized. We illustrate the cases of three SCD pregnant patients who underwent aRBCX procedures at our institution in different clinical scenarios. Moreover, we carried out a careful literature revision to investigate the management of pregnancy in SCD, with a particular focus on the viability of aRBCX. Our experience and the current literature support the use of aRBCX in pregnancy as a feasible and safe procedure, provided that specialized equipment and an experienced apheresis team is available. However, further research in this high-risk population, with appropriately powered prospective trials, is desirable to refine the indications and timing of aRBCX and to confirm the advantages of this approach on other transfusion modalities.

## 1. Introduction

Sickle cell disease (SCD) is an inherited hemoglobinopathy characterized by the presence of abnormal hemoglobin, resulting in the formation of hard and sticky C-shaped red cells [1]. This is caused by a single base pair point mutation in the sixth position of the beta globin chain, leading to the substitution of glutamic acid by valine (GAG to GTG). SCD encompasses homozygous mutations (S/S) as well as combined hemoglobinopathies, which may have a similar clinical presentation, such as the heterozygous mutation with a β0-thalassemia (S/β0) or other mutations (e.g., HbSC) [2,3]. SCD results in the formation of abnormal hemoglobin polymers when deoxygenated, with decreased deformability and the typical sickle form of the erythrocytes. The clinical manifestations of SCD are varied, due to the short life span of sickled red blood cells and their tendency to become stuck in the blood vessels [4]. In fact, sickled erythrocytes possess many unfavorable physiologic properties and induce vascular changes that promote vaso-occlusion, infarction, hemolysis and inflammation. Clinical manifestations of SCD are vaso-occlusive (VOC) crisis, acute pain syndrome, organ infarction and hemolytic anemia. Relapsing vaso-occlusive crises can affect multiple organ systems, and SCD patients have an increased risk of stroke, renal dysfunction, pulmonary hypertension, retinal illness and avascular necrosis [1,2,3,4,5].

Currently, the availability of proper care and appropriate treatment strategies, including the wide use of hydroxyurea as inducer of fetal hemoglobin synthesis, regular transfusions and iron chelation therapy, have resulted in increased life expectancy and quality of life in patients suffering from SCD [6,7]. The prolonged life span also increases the possibility for SCD patients to undergo pregnancies. Pregnancy in SCD is a high-risk situation, especially during the third trimester of gestation and in the post-partum period, due to chronic hypoxia and vaso-occlusive phenomena in the maternal–fetal microcirculation [8]. Of note, an increased incidence of abortion is reported, related to an increased risk of pre-eclampsia, preterm deliveries and pulmonary embolism. In addition, fetal complications, such as intra-uterine growth restriction, prematurity or fetal loss are more frequent in SCD pregnancies [9]. Hydroxyurea, commonly used in symptomatic patients, is contraindicated in pregnancy due to concerns for the fetus health. Indeed, transfusion therapy is the mainstay to relieve SCD symptoms. Nonetheless, the transfusion management of pregnant SCD patients remains controversial; although there is a consensus on the need for a strict and multidisciplinary follow-up within specialized structures, the benefit of prophylactic management, either by simple transfusions or by automated red blood cell exchange (aRBCX), is not unanimously recognized [10,11,12].

In this study, we illustrate the cases of three pregnant SCD patients who underwent aRBCX procedures in different clinical situations. Moreover, we reviewed the current literature to investigate the management of pregnancy in SCD, particularly regarding the practice of aRBCX.

## 2. Materials and Methods

### 2.1. Patients, Treatment and Procedures

Three case reports are reported of aRBCX performed either for prophylactic or therapeutic indications. Data were anonymously recorded, and patients’ informed consent was obtained. According to local procedures, all SCD patients were serologically typed for ABO, Rh (C, D, E, c, e), Kell, Duffy (Fya, Fyb), Kidd (Jka, Jkb) and MNS (S, s, M, N) blood group antigens. In addition, extended RBC antigen typing was determined and confirmed by molecular biology techniques using human erythrocyte antigen (HEA) BeadChip (BioArray Solutions Ltd., Warren, NJ, USA) [13]. Before transfusion, the irregular antibody screening was performed and red blood cell (RBC) units were selected according to the broadest possible donor–recipient match. Prestorage leukoreduced RBCs with an Hct of approximatively 60% were used. Transfusion therapy was administered to maintain Hb levels between 9 and 10 g/dL. RBC units were transfused with or without concomitant phlebotomy, depending on the pre-transfusion hematocrit (Hct) value: for Hct < 30%, no phlebotomy was carried out, whereas for Hct ≥ 30%, patients usually received infusion of saline solution, isovolemic phlebotomy and then RBC unit transfusion. The aRBCX was performed using the Optia cell separator (Optia Spectra Apheresis System; Terumo BCT, Lakewood, CO, USA). Pre- and post-RBCX complete blood counts and HbS percentage (HPLC, BIO-RAD, Hercules, CA, USA) were regularly determined. The exchange volumes were estimated according to patient gender, age, body weight, height, pre-procedure Hct and HbS level and average Hct of blood products. The fraction of residual RBCs (FCR) and the volume to be replaced were estimated to achieve a target HbS of <30% and a final Hct of 30 ± 3% to avoid hyperviscosity [14,15]. Peripheral venous accesses were used whenever possible; otherwise, a temporary central vascular catheter (CVC) was placed. Data about single aRBCX procedure in pregnancy (venous access, blood volume substituted, HbS target and desirable FCR) were collected.

### 2.2. Systematic Review

We conducted a systematic review on red blood cell exchange in the management of SCD in pregnancy. The reporting of this systematic review was guided by the standards of the Preferred Reporting Items for Systematic Review and Meta-Analysis (PRISMA) statement, when applicable. We performed a systematic search on PubMed database using the following queries: “Pregnancy” [Mesh] AND “Anemia, Sickle Cell” [Mesh] AND “Blood Component Removal” [Mesh]. No additional search filters were applied. C.G.V., C.P., S.C. and L.T. independently controlled all references, including case reports, case series and reviews. Discrepancies were discussed by the authors and resolved. Papers not reporting procedures of erythrocytapheresis in pregnant homozygous HbS women, communications at congresses, duplicated studies, case reports, narrative reviews and papers with an abstract in languages other than English were excluded. Up to August 2023, 11 total references were identified. We additionally screened the reference lists and the first 20 “similar articles” in PubMed of included studies for additional eligible studies. In the end, seven papers were included and discussed in the review (Figure 1). Collected data included study type, patient characteristics, erythrocytapheresis schedule and indications, procedure’s technical details, maternal and fetal outcomes. The overall quality of evidence related to maternal and fetal outcomes was assessed using the Grading of Recommendations Assessment, Development and Evaluation (GRADE) approach, which considers factors such as study design, risk of bias (limitations in study design), inconsistency in results, indirectness of evidence, the precision of effect estimates, publication bias, the magnitude of effect, dose–response gradient and all possible confounding factors [16,17].

## 3. Results

In Table 1, we report the main features of SCD patients described as follows, and the technical details of apheresis procedures performed in pregnancies. 

### 3.1. Clinical Reports

#### 3.1.1. Case 1

A 37 year-old patient was affected by sickle cell anemia combined with the 3.7 deletion of the hemoglobin alpha gene. She irregularly presented herself at hematological controls and was not on a regular transfusion program. The co-inheritance of SCD with α 3.7 deletion is associated to a less severe SCD phenotype and it could explain her stable baseline Hb level, decreasing the tendency of HbS to polymerize and reducing the rate of hemolysis [18]. She had received transfusions only on the occasions of her previous three pregnancies in 2016 and 2017, all exited in miscarriages at gestational age of 10, 14 and 15 weeks, respectively. She was also subjected to two aRBCX courses in emergency: in 2018 for a VOC with severe hemolysis and in 2019 for a painful crisis due to gallbladder stones. In 2021, she resumed hematological controls since she was 14 weeks into her fourth gestation. Patient history was silent for recent painful crises, VOCs or thromboembolic events. Antithrombotic prophylaxis was promptly set up with enoxaparin 4000 U/die in addition to cardioaspirin 100 mg/die; meanwhile a transfusion support every three weeks was resumed, in association with phlebotomy to maintain Hct within 30%. Obstetric echo tomography on the second and third trimesters were unremarkable. The HbS level during pregnancy ranged between 60% and 70%. The patient was completely asymptomatic until week 36, when she presented herself at our emergency department with fever and diffused pain to the whole vertebral region and was poorly responsive to the antalgic therapy. The search for SARS-CoV-2 infection was positive. An aRBCX was performed, reducing the HbS level from 65% to 29%. Clinical conditions progressively improved and the patient was discharged (Table 1, Procedure 1A). Pregnancy ended 2 weeks later with a caesarean section for non-response to birth induction, with no maternal or fetal complications. Histological examination of placenta and fetal annexes showed maternal vascular malperfusion and intravillar thrombosis (Figure 2).

#### 3.1.2. Case 2

The patient’s diagnosis dates back to 1994, when she was 12 years old, with the onset of thrombophlebitis of the left leg, and since then she had been on a regular blood transfusion regimen at different hospitals. The patient was lately referred to our hematology department in 2011, at the age of 23, during her first pregnancy; a silent medical history was collected at that time, with the exception of two previous surgeries (ovarian cysts removal in 2008, and tonsillectomy in 2010) performed without complications. She was initially treated with monthly simple transfusion to maintain Hb levels between 9 and 10 g/dL. During the second trimester of pregnancy, the patient experienced widespread lumbar pain and dyspnea due to a severe hemolytic crisis, requiring the first aRBCX procedure with rapid clinical benefits (Table 1, Procedure 2A). The patient continued transfusion support; at week 35, an urgent caesarean section was performed for amnionitis: no adverse maternal or fetal outcomes were recorded. During the following months, the patient experienced recurrent VOCs, so she stopped breastfeeding and started therapy with hydroxyurea, discontinued in 2015 for intolerance and occurrence of skin ulcers; the patient then maintained transfusion support on a monthly basis. In 2016 and 2018, she experienced two other pregnancies, both resulting in miscarriages at approximatively 8 weeks; no antithrombotic prophylaxis or RBC exchange schedule were implemented due to the early occurrence of the adverse obstetric events. In 2019, the patient had her fourth pregnancy: antithrombotic prophylaxis with enoxaparin was promptly started and transfusion regimen was intensified, in association with phlebotomies to maintain an average Hb value between 9 and 10 g/dL and an Hct lower than 30%. The patient frequently complained of discomfort from phlebotomies. In addition, she referred to occasional painful episodes, responsive to anti-inflammatory oral therapy. To limit the antalgic therapy requirement, an initial aRBCX procedure was performed at week 16 of gestation, lowering the HbS values from 46% to 19% (Table 1, procedure 2B). A second aRBCX was prophylactically performed at week 28, with an HbS reduction from 40% to 25% (Table 1, Procedure 2C). Pregnancy evolved without fetal distress or intrauterine growth retardation as documented by serial echotomography assessment, which only showed a slight increase in the pulsatility index (PI) of the uterine arteries at week 20, not further confirmed at subsequent controls. The patient delivered at week 36 + 4 with an elective caesarean section without gynecological and fetal complications. Histological examination of placenta and fetal annexes was performed, with findings of maternal vascular malperfusion, perivillar fibrin deposits and evidence of drepanocytes in the intervillous spaces (Figure 3).

#### 3.1.3. Case 3

The patient was diagnosed with sickle cell anemia at the age of 6. She was first evaluated at our center in 2014, at the age of 34, during her first pregnancy; until then, she had been scarcely symptomatic, and she seldom received transfusions. At the time of evaluation, she was at the 8th week of gestation: antiplatelet prophylaxis with a low dose of molecular heparin was settled, and the patient remained asymptomatic without need for transfusions while pregnancy regularly progressed for the first two trimesters. At week 24 of gestation, a prophylactic aRBCX was performed (Table 1, Procedure 3A). At week 38, the patient underwent a planned caesarean section and delivered a healthy newborn. The patient returned to our observation in 2021 for a new initial pregnancy (week 7). She reported two abortion episodes at 11 and 10 weeks of gestation, respectively, both occurring under antiplatelet therapy. Considering the HbS value of 90% and the previous obstetrical history, antithrombotic prophylaxis with enoxaparin was started, and an aRBCX was promptly performed (week 8 of gestation, Table 1, Procedure 3B). The HbS level was reduced to 21%; a regular transfusion support every three weeks was then started, with occasional phlebotomies depending on the Hct level. Due to the progressive raise in HbS levels (76%), a new aRBCX was performed at week 34 of gestation, resulting in the HbS decrease to 31% (Table 1, Procedure 3C). Then, the patient underwent an elective caesarean section at week 39; no gynecological or fetal complications were recorded.

### 3.2. Literature Revision

In total, eleven studies were retrieved from the literature search. Seven articles were excluded after abstract and full text screening. Three additional papers were identified and included after verifying the reference lists and the first 20 “similar articles” in PubMed of the already included studies. Seven papers were finally discussed in the review. The results are summarized in Table 2 [19,20,21,22,23,24,25]. Overall, the level of evidence was deemed very low.

## 4. Discussion

Appropriate transfusion support for patients with SCD is still a matter of debate and the use of erythrocytapheresis rather than simple transfusions during pregnancy remains controversial [8]. The literature review indicates that therapeutical approaches are very different among specialist centers. All the studies included in this review, mostly retrospective, investigated the feasibility and the safety of aRBCX during pregnancy, exploiting the association between erythrocytapheresis-based approaches and maternal and fetal outcome [19,20,21,22,23,24,25]. However, it is difficult to achieve conclusive clinical indications, due to the heterogeneity of apheresis schedules, the timing of RBCX procedures and the different clinical outcomes evaluated.

The American Society for Hematology and British Society of Hematology guidelines for management of SCD in pregnancy do not recommend to routinely start a program of prophylactic transfusion, but to consider the indication on a case-by-case basis, especially in women with a history of severe SCD-related complications before the current pregnancy [10,11]. RBC transfusion should be considered a standard of care treatment, if and when clinically indicated on the basis of hemoglobin value and HbS level [12,26], while ASFA guidelines for therapeutic apheresis recommend aRBCX as a second-line treatment in pregnant patients (Category II, Grade 2B) [27].

The most frequent therapeutic indication for RBCX is acute stroke, while it is indicated as prophylactic measure to prevent the recurrence of cerebrovascular accidents and before extensive surgery [26,27]; likewise, only a few papers, mostly retrospective studies [19,20,21,22,23,24,25] with sometimes a control group [21,22,23,25] or sporadic cases reports [28,29], address the use of RBCX in pregnancy, without reaching reliable conclusions.

Globally, it is estimated that there are over 300,000 births per year of individuals with SCD, and over 75% of these are in sub-Saharan Africa [30]. Automated RBCX is a specialized service, and unfortunately, due to the scarce resources, its diffusion is limited in Africa and in other low- and middle- income countries. However, efforts in recent years to incorporate at least manual exchange blood transfusion into routine care are ongoing in some countries of sub-Saharan Africa [31]. In Tanzania, for example, manual exchange blood transfusion was first reported in 2017. Since then, it has been performed in both public and private tertiary health facilities in patients with either acute or chronic complications [31,32], but no cases of pregnancy women with SCD are reported.

The first description of erythrocytapheresis for sickle cell disease during pregnancy dates to 1980, when Key et al. reported outcomes on eight pregnant women: all pregnancies were carried to term, with the delivery of healthy infants. There was no fetal or neonatal morbidity, except for a case of puerperal endometritis [19]. Later on, Lee et al., in 1991, investigated possible RBCX-induced hemodynamic modifications occurring either in the mother or in the fetus: they observed only a slight tendency for reduced post-RBCX cardiac output in the mother, while no alterations were detected in the fetus. The authors concluded that changes in maternal hemodynamic and metabolic functions were negligible [20]. In our experience, given the fragility of SCD patients, both the transfusions and the apheretic procedures of aRBCX were performed safely, in a hospital environment. Given the lack of formal evidence to guide clinical practice, we believe that pregnant women with SCD should be managed by a multidisciplinary team including hematologists and obstetricians with experience in high-risk pregnancies. Close collaboration between hematologists and gynecologists is necessary to monitor the progress of the pregnancy and the growth of the fetus to intercept any early signs of fetal anemia or growth retardation.

Among the cases described above, the aRBCX was adopted as a “prophylactic approach” to prevent obstetric complications in all the three patients. Notably, all of them had undergone periodic ultrasound monitoring of fetal growth with uterine and cerebral Doppler velocimetry, not showing fetal distress or intrauterine growth retardation, as documented by serial echotomography assessment. Nevertheless, we documented microcirculation obstructions at placental histopathologic findings with the presence of perivillar fibrin deposits, villar hypoplasia and parenchymal infarctions, all signs of maternal vascular malperfusion [33]. Of note, a patient experienced aRBCX early in the gestation, at week 8 of gestation. Maternal vascular malperfusion is the predominant lesion in placentas of women with SCD and is strongly associated with adverse pregnancy outcomes, mainly small for gestational-age infants, preterm birth and stillbirth [34]. The same histologic findings were documented by Vianello et al., who conducted a retrospective study enrolling 18 SCD women in a program of early prophylactic erythrocytapheresis in association to LMWH [24]: aRBCX was carried out every 3 or 4 weeks during pregnancy starting from a mean of 10.7 weeks of gestation until delivery, for a total of 160 procedures. The authors reported a positive impact on maternal outcomes (no episodes of severe VOCs, acute chest syndrome and eclampsia were observed) and an improvement in newborn birthweight compared with previous studies. However, 6.5% of the pregnancies resulted in stillbirth, and placental histopathological examination showed signs of maternal vascular malperfusion and erythroblastosis in cord blood, compatible with fatal hypoxia due to vascular insufficiency [24]. All these findings suggest that, even despite the absence of manifest obstetrical complications the pro-oxidant and pro-inflammatory environment sustained by chronic hemolysis and vaso-occlusion, with cycles of ischemia and reperfusion, inevitably causes a placental impairment. Accordingly, transcriptome analyses highlighted in placenta samples from SCD pregnancies the altered expression of several genes associated with migration, trophoblast differentiation and inflammation [35]. Interestingly, recent observations in patients with SCD report abnormalities in placental DNA methylation status, leading to gene expression changes also in cases without evident clinical impact [36].

The effects of prophylactic aRBCX procedures on a considerable group of patients were first reported by Morrison et al., who described 131 pregnant women who received apheresis during pregnancy, while the control group received simple transfusion support [21]. Patients on a regular aRBCX regimen had a lower incidence of pregnancy-related complications (prenatal death, low birthweight infants and preterm deliveries) and a reduced hospital stay when compared to the control group. Two patients developed post-transfusion hepatitis and five had post-transfusion reactions [21]. Similarly, a lower risk of intrauterine growth restriction and oligohydramnios in pregnant women treated with aRBCX was also reported by other authors, who investigated the effects of prophylactic transfusion by means of erythrocytapheresis at the beginning of the third trimester of pregnancy, suggesting a potential improvement in fetal morbidity [22]. Asma et al. retrospectively evaluated the complications of SCD in 37 pregnant patients; 24 patients received 43 prophylactic exchange procedures at variable time points, and they were compared with a control group of 13 patients [23]. There was a significant difference in maternal mortality and incidence of VOCs between the study and control groups; however, due to study limitations, no difference in fetal complications’ incidence was stated [23]. In a subsequent study performed at the same institution, a higher rate of painful crises, preeclampsia and preterm birth in the control vs. prophylactic aRBCX group were reported [25]. A meta-analysis performed by Malinowski et al. in 2015 assessed the effects of prophylactic compared with on-demand red blood cell transfusions on maternal outcomes in pregnant SCD women [37]; three studies involving prophylactic RBCX were also included. The authors concluded that this approach may positively affect maternal complications and reduce perinatal mortality, but these results were weakened by the paucity and the low quality of the available evidence [37]. No other study has since been published comparing prophylactic aRBCX to other treatment modalities in SCD patients.

Apheresis-related adverse events may include a wide range of symptoms: the most frequent are those secondary to citrate effect (paresthesia, nausea) or venous access difficulty, and are usually mild or moderate. However, severe episodes of allergic reactions, hypotension, hypertension, respiratory distress and chest pain have also been reported [38]. In a recent study revising a large number of procedures, aRBCX seems to be associated with lower adverse event incidence in comparison with plasmapheresis [39]. Noteworthy, no maternal or fetus aRBCX-related adverse events have been reported in the above-mentioned studies exploring aRBCX in SCD and pregnant patients. Evidently, the limited number of studies so far conducted, and the very low quality of the evidence, do not make it possible to provide strong recommendations about a regular prophylactic use of aRBCX. Currently, a multicenter feasibility trial to evaluate serial prophylactic exchange blood transfusion in pregnant women with SCD (TAPS-2, NCT03975894) is ongoing in the UK [40].

The main aim of the present study was to provide a revision of the literature regarding aRBCX use in the management of SCD and pregnant patients. From this revision, it clearly emerges that robust indications on the prophylactic use of aRBCX are lacking. For this reason, aRBCX practice may differ among centers depending on the availability and accessibility to apheresis, or even on the previous center’s experience, supporting or not aRBCX reliability in the prevention of obstetric complications. Regarding our center, we experienced no adverse events in SCD patients treated by either prophylactic or therapeutic aRBCX. Presently, we tend to propose aRBCX to all SCD and pregnant patients who faced previous obstetrical complications while in a regular transfusion program. Usually, we plan two procedures, one during the last part of the first trimester and the other one at the end of the second trimester. We continue RBC transfusions after the aRBCX procedure to limit the rapid rise in HbS levels. We acknowledge that, among aRBCX procedures performed at our hospital on SCD patients, only a minority is performed on pregnant patients. This prevents us to contribute large-case series or to generalize our approach. Notwithstanding this limitation, however, we perceive aRBCX as a safe procedure to give SCD patients better chances to cope with a condition with potential harms for the fetus and the mother [8,9].

## 5. Conclusions

Prophylactic aRBCX may be a useful and safe tool for the management of pregnancy in SDC patients who experienced previous obstetrical complications. The drawbacks of this approach reside in the need for specialized equipment and experienced apheresis teams, with a limited access for patients in low-income countries, where SCD is highly prevalent. Appropriately powered prospective trials are warranted to further explore the role of aRBCX in this high-risk group of pregnant women.

## Figures and Tables

**Figure 1 jcm-12-07123-f001:**
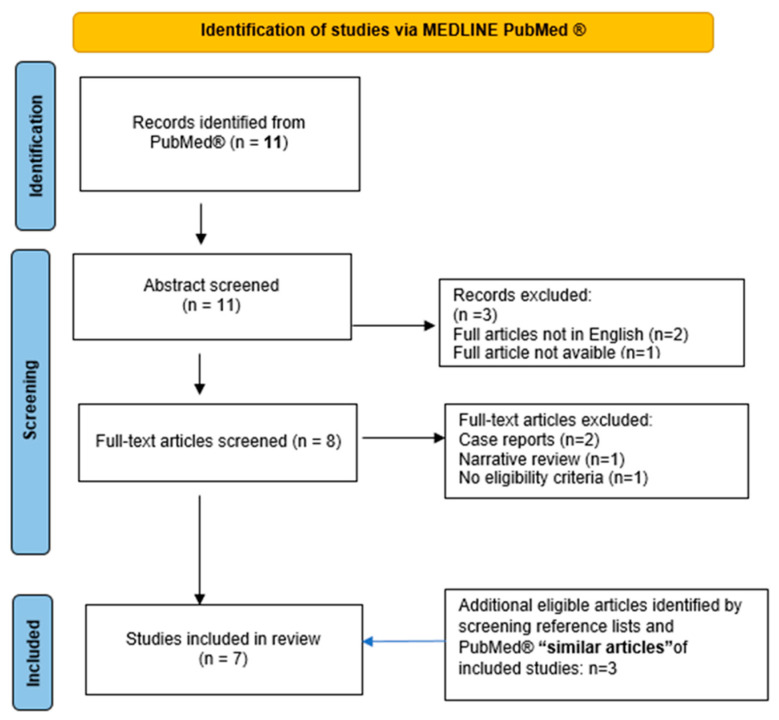
Flowchart for study selection.

**Figure 2 jcm-12-07123-f002:**
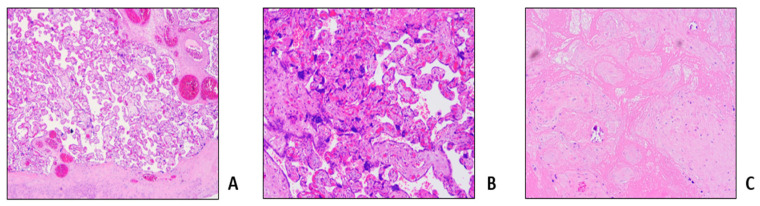
Microscopic findings in a 39-week placenta (weight 552 g). Some areas of delayed villous maturation (**A**) along with areas of increasing in syncytial knots (**B**) and parenchymal infarctions (**C**).

**Figure 3 jcm-12-07123-f003:**
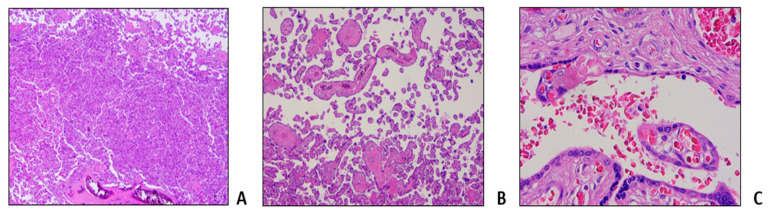
Microscopic findings in a hypoplasic 38w placenta (weight 472 g). Villar agglutination (**A**) along with areas of villar hypoplasia (**B**). These histopathological findings are classified as maternal vascular malperfusion. Maternal drepanocytes in the intervillous space (**C**).

**Table 1 jcm-12-07123-t001:** Main features of patients and apheresis procedures performed during pregnancy.

	Case 1	Case 2			Case 3		
Genotype	HbS c.20A>T + trait alfa-thalassemia (del alfa 3.7)	HbS c.20A>T			HbS c.20A>T		
Blood group and red cell phenotype	A POS ccee kk	AB pos ccee kk Fya-b-			0 pos ccee kk Fya+b-		
	Fya-b- Jka+b- Kpa-b+	Jka+b- Kpa-b+			Jka+b- Kpa-b+		
	Jsa-b+ Ns Lua-b+ Dia-b+	Jsa-b+ MNs Lua-b+ Dia-b+			Jsa-b+ MNs Lua-b+ Dia-b+		
	Doa+b+ Coa+b- Sc 1,2	Doa-b+ Coa+b- Sc 1,2			Doa+b+ Coa+b- Sc 1,2		
Obstetric history (n) ✓ Pregnancies ✓ At term ✓ Miscarriage (gestational age, weeks)							
	4	4			4		
	1	2			2		
	3 (10, 14, 15)	2 (8, 8)			2 (11, 10)		
At term pregnancy	First	First	Second		First	Second	
Antithrombotic prophylaxis	Cardioaspirin 100 mg/die + Enoxaparin 4000 U/die from 14 weeks to delivery	None	Enoxaparin 4000 U/die from 12 weeks to delivery		Enoxaparin 4000 U/die from 25 weeks to delivery	Enoxaparin 4000 U/die from the beginning to delivery	
Median transfusion requirement pre-apheresis (RBC units/month)	1	1	1		1	0	
	**Procedure 1A**	**Procedure 2A**	**Procedure 2B**	**Procedure 2C**	**Procedure 3A**	**Procedure 3B**	**Procedure 3C**
aRBCX indication	Vaso-occlusive crisis	Vaso-occlusive crisis	Prophylactic	Prophylactic	Prophylactic	Prophylactic	Prophylactic
Gestational age at procedure (weeks)	38 + 2	19 + 5	16 + 3	28 + 1	24 + 0	7 + 5	34 + 0
Weight (kg)	80	68	68	75	51	49	50
Pre-treatment HbS (%)	65	64.7	46	40	45.4	87	47
Vascular access	Peripheral	Temporary CVC	Peripheral	Peripheral	Temporary CVC	Peripheral	Peripheral
Post-treatment HbS (%)	29	7.8	19	25	27	21	25
Blood volume substituted (mL)	4164	4796	4930	3276	2769	4497	2844
Target FCR (%)	26	28	38	49	39	19	39
Obstetric outcome: ✓ Birthweight	C-section at week 39 + 0	C-section at week 35 + 1	C-section at week 36 + 4		C-section at week 37 + 3	C-section at week 39 + 2	

Legend: RBC, red blood cell; aRBCX, automated red blood cell exchange; FCR, fraction of cells remaining; CVC, central vascular catheter.

**Table 2 jcm-12-07123-t002:** Studies reporting therapeutic approach with red blood cell exchange in patients with sickle cell disease and pregnancy.

Reference	Study Type	Population/Pregnancies (n)	Apheresis Sessions Schedule (n = Pregnancies)	Total Procedures (n)	Apheresis System	Technical Details	Vascular Access	Outcome	Level of Evidence (Quality Assessment [17])
Key et al., 1980 [19]	Single-center, retrospective	8/8	First prophylactic RBCX at variable gestational age (range 17–30 weeks); 2 patients underwent a second procedure for HbA < 25%	10	IBM COBE 2997 Blood Cell Separator	Not reported	Peripheral	All pregnancies were carried to term. No fetal or neonatal morbidity.	VERY LOW (⨁)
Lee et al., 1990 [20]	Single-center, prospective	5/5	Prophylactic RBCX during second or early third trimester when Hct < 25%, and HbS > 65%	5	IBM COBE 2997 Blood Cell Separator	Not reported	Not reported	Significant increases in the Hct and % HbA. Negligible changes in maternal hemodynamic and metabolic function.	NA
Morrison et al., 1990 [21]	Single-center, retrospective	131/131	Prophylactic RBCX (n = 103): first procedure as early in pregnancy as possible, then if HbA < 20% or severe crisis or morbidity. Control group (n = 28): simple transfusion support.	Not reported	IBM COBE 2997 Blood Cell Separator	HbA target > 50%	Not reported	Lower maternal morbidity rates and hospitalization days. Decreased number of preterm deliveries, decreased prevalence of low birthweight infants and perinatal death rate.	VERY LOW (⨁)
Gilli et al., 2007 [22]	Single-center, retrospective	31/31	Prophylactic RBCX from the 28th week onwards (n = 14). Control group (n = 17): simple prophylactic transfusions.	Not stated	Not reported	Not reported	Not reported	Lower risk of intrauterine growth restriction and oligohydramnios.	VERY LOW (⨁)
Asma et al., 2015 [23]	Single-center, retrospective, cross-sectional	37/37	Prophylactic RBCX (n = 24): 1–3 at variable pregnancy time points. Control group (n = 13): simple transfusion support.	43	Cobe Spectra 7.0, Spectra Optia 7.0	Hbs target < 30% FCR: 60–70%	Not reported	Higher rates of maternal mortality, maternal complications, incidence of VOC crises and fetal complications in control group.	VERY LOW (⨁)
Vianello et al., 2018 [24]	Double-center retrospective cross-sectional study	18/46	Every 3–4 weeks during pregnancy until delivery, starting at variable pregnancy time points (range from 22 to 28 weeks).	160	COM.TEC, Fresenius Kabi	Hbs target <30%	157 peripheral (98.1%), 3 temporary CVC (1.9%)	No severe VOCs, sepsis, severe infection. Normal umbilical artery impedance during pregnancy. Improvement in new-born birthweight compared to mean values of SCD pregnancies reported in literature.	VERY LOW (⨁)
Baran et al., 2021 [25]	Single-center, retrospective, cross-sectional	37/46	Prophylactic RBCX (n = 27): 1 or more sessions at variable pregnancy time points (25–30 weeks). Therapeutic RBCX (n = 7): severe VOCs. Control group (n = 19): simple transfusion support	43	Spectra Optia 7.0	Hbs target < 30% FCR: 60–70%	20 peripheral (60.6%), 13 temporary CVC (39.4%)	Higher rate of painful crises, preeclampsia and preterm birth in control vs. prophylactic RBCX group.	VERY LOW (⨁)

Legend: SCD, sickle cell disease; VOC, vaso-occlusive crisis; RBC, red blood cell; aRBCX, automated red blood cell exchange; FCR, fraction of cells remaining; CVC, central vascular catheter; NA, not applicable.

## Data Availability

Due to the confidentiality agreements, the data analyzed for this study are only available from the corresponding author upon reasonable request.

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
