# Peer review of "Red Blood Cell Exchange as a Valid Therapeutic Approach for Pregnancy Management in Sickle Cell Disease: Three Explicative Cases and Systematic Review of Literature"

_jcm, 2023, doi:10.3390/jcm12227123_

Round 1
Reviewer 1 Report
Comments and Suggestions for Authors
I think this is a narative review regarding the outcome of pregnancy in patients with sickle cell anemia and no validation of erythrocytapheresis could be extracted from the review.
page 213 drepanocytosis: I think it is not the actual term. It is better to switch sickle cell anemia
You could clarify when we start acetylsalicylic acid in a pregnant with SCA and what dose. When we start low molecular weight heparin You could write in discussion that the most published data are retrospective , with or without control.
At what hemoglobin level do you start transfusion?
You could discuss more the findings of placenta and give some suggestions.
You could report whether erythrocytapheresis is available in Africa and in countries with low incomes or underdeveloped. What heppened there?
Author Response
We appreciated the positive comment of Reviewers.
We are grateful to the referees and to the Editorial staff for your efforts in providing the review, which significantly improved the quality of our paper.
We read with great attention all the stimulating suggestions and accordingly, we modified our manuscript, following each point raised and highlighting in yellow the corrections in the text. In addition, we made an extensive English revision of our paper checked by a native English speaker.
- REVIEWER 1:
- page 213 drepanocytosis: I think it is not the actual term. It is better to switch sickle cell anemia.
Reply. As suggested, we substituted the term drepanocytosis with sickle cell anemia (line 222).
- You could clarify when we start acetylsalicylic acid in a pregnant with SCA and what dose. When we start low molecular weight heparin. You could write in discussion that the most published data are retrospective, with or without control.
Reply. The dose and the gestational age of starting cardioaspirin and low molecular weight heparin are indicated in the table 1, and we added the dosages in the text, too (lines 162, and 164-165). As for the revision of literature, we highlighted that the published studies are mostly retrospective, sometimes with a control group (lines 255, and 271-273).
- At what hemoglobin level do you start transfusion?
Reply. Transfusion therapy was administered to maintain Hb levels between 9 and 10 gr/dl (lines 85-86).
- You could discuss more the findings of placenta and give some suggestions.
Reply. As suggested, we further expanded in the Discussion the paragraph on the histopathological placental findings observed in the present study (lines 319-327). According with recent evidences, we enforced our suggestion that placental abnormalities occur also in absence of evident clinical impact. For this purpose, we added to the reference list 3 new citations (ref. n 33,35,36).
- You could report whether erythrocytapheresis is available in Africa and in countries with low incomes or underdeveloped. What happened there?
Reply. The use of aRBCX is limited in Africa and in other developing countries. However, efforts to incorporate manual exchange blood transfusion into routine care is ongoing. In Tanzania, manual exchange blood transfusion was first reported in 2017, and since then it has been performed in both public and private tertiary health facilities in patients with either acute or chronic complications. As suggested, we added this paragraph in the text (lines 274-282), with 3 new references (ref. n 30,31,32).
We thank you for having given us the opportunity of revising the paper according to the referee’s comments.
Hoping that the revised version of our work might be suitable for publication in Journal of Clinical Medicine, I send my best regards.
Sincerely yours,
Caterina Giovanna Valentini
Transfusion Medicine Department
Fondazione Policlinico Universitario A. Gemelli IRCCS
I-00168 Roma, Italy
39-06-30154180
caterinagiovanna.valentini@policlinicogemelli.it
Reviewer 2 Report
Comments and Suggestions for Authors
The manuscript is well written. I have a few questions/concerns:
1. Please correct all the grammatical and syntactical errors that exist throughout the manuscript and have it checked by a native english speaker, experienced in medical writing. Please provide the relevant certificate.
2. As an overall conclusion, is RBC Exchange in women with SCD dangerous for the foetus or does it help the foetus and the pregnant woman? Please add a novel paragraph in the discussion making it clear.
3. What are the restrictions of your study? Please write a novel pragraph in the discussion addressing this.
Comments on the Quality of English LanguageThe manuscript needs language polishing.
Author Response
We appreciated the positive comment of Reviewers.
We are grateful to the referees and to the Editorial staff for your efforts in providing the review, which significantly improved the quality of our paper.
We read with great attention all the stimulating suggestions and accordingly, we modified our manuscript, following each point raised and highlighting in yellow the corrections in the text. In addition, we made an extensive English revision of our paper checked by a native English speaker.
- REVIEWER 2:
- Please correct all the grammatical and syntactical errors that exist throughout the manuscript and have it checked by a native english speaker, experienced in medical writing. Please provide the relevant certificate.
Reply. As suggested, we carefully revised the manuscript for the English language by a native English speaker, experienced in medical writing.
- As an overall conclusion, is RBC Exchange in women with SCD dangerous for the foetus or does it help the foetus and the pregnant woman? Please add a novel paragraph in the discussion making it clear.
Reply. As suggested, we added to the Discussion a brief paragraph on the safety of RBC Exchange in general and in the specific setting of SCD pregnancy (lines 353-362), adding to the reference list 2 new citations (ref. n 38 and 39).
- What are the restrictions of your study? Please write a novel paragraph in the discussion addressing this.
Reply. We added at the end of the Discussion a new paragraph reporting that the approach we described derives from our local experience and it cannot be generalized (lines 365-380). Moreover, in the Conclusion we clearly stated that it is scarcely applicable in low-income countries where SCD is highly prevalent (lines 384-389).
We thank you for having given us the opportunity of revising the paper according to the referee’s comments.
Hoping that the revised version of our work might be suitable for publication in Journal of Clinical Medicine, I send my best regards.
Sincerely yours,
Caterina Giovanna Valentini
Transfusion Medicine Department
Fondazione Policlinico Universitario A. Gemelli IRCCS
I-00168 Roma, Italy
39-06-30154180
caterinagiovanna.valentini@policlinicogemelli.it
Reviewer 3 Report
Comments and Suggestions for Authors
The study describes three cases of SCD patients who present with pregnancy, as well as a review of the literature concerning the blood transfusion approach during a sickle cell pregnancy. The latter is really useful because the subject is complex and deserves an update.
The three cases illustrate the difficulty of managing a complicated pregnancy in a sickle cell patient and this even in the case of a patient already in a transfusion program. However, this only reports cases. An assessment of the approach carried out such as the place of blood transfusion, the multidisciplinary approach, the RBC exchange risk for these patients, etc., would be worth discussing.
For the literature review, the “level of evidence” aspect should be added.
It would be interesting to give a "specification" of what remains to be done, or not, to be able to give recommendations on 1) the place of blood transfusion, 2) the blood transfusion modalities (aRBCX), in sickle cell pregnancy.
In the conclusion, it is described that the authors experience demonstrates the efficacy of aRBCX at early SCD pregnancy to reduce HbS level. Unless misunderstanding, no results were given in the article (how many SCD patients followed, etc.).
Author Response
We appreciated the positive comment of Reviewers.
We are grateful to the referees and to the Editorial staff for your efforts in providing the review, which significantly improved the quality of our paper.
We read with great attention all the stimulating suggestions and accordingly, we modified our manuscript, following each point raised and highlighting in yellow the corrections in the text. In addition, we made an extensive English revision of our paper checked by a native English speaker.
- REVIEWER 3:
- The three cases illustrate the difficulty of managing a complicated pregnancy in a sickle cell patient and this even in the case of a patient already in a transfusion program. However, this only reports cases. An assessment of the approach carried out such as the place of blood transfusion, the multidisciplinary approach, the RBC exchange risk for these patients, etc., would be worth discussing.
Reply. We agree with Reviewer suggestion. The place where pregnant patients receive the transfusion varies, depends on whether the patients were outpatients or hospitalized. In all cases the transfusions were performed safely, in a hospital environment, such as the apheretic procedures of aRBCX. Given the lack of formal evidence to guide clinical practice, we believe that pregnant women with SCD should be managed by a multidisciplinary team including hematologists and obstetricians with experience in high-risk pregnancy. The optimal approach in managing pregnancy complicated by SCD, that results in improved maternal and fetal outcomes, is yet to be identified. Close collaboration between hematologists and gynecologists is necessary to monitor the progress of the pregnancy, and the growth of the fetus, to intercept any early signs of fetal anemia or growth retardation. We added this consideration in the text (lines 291-298). In addition, we add a paragraph on the aRBCX side effects and risks (lines 253-262), with 2 new references (ref. n 38 and 39).
- For the literature review, the “level of evidence” aspect should be added. It would be interesting to give a "specification" of what remains to be done, or not, to be able to give recommendations on 1) the place of blood transfusion, 2) the blood transfusion modalities (aRBCX), in sickle cell pregnancy.
Reply. We provided the level of evidence of the revised manuscripts, using the Grading of Recommendations Assessment, Development, and Evaluation (GRADE) approach. Accordingly, we modified the paragraph 2.2 Systematic review, and provided the level of evidence in Table 2. Moreover, 2 citations were added (ref. n 16 and 17).
- In the conclusion, it is described that the authors experience demonstrates the efficacy of aRBCX at early SCD pregnancy to reduce HbS level. Unless misunderstanding, no results were given in the article (how many SCD patients followed, etc.).
Reply. We agree with the Reviewer criticism. Accordingly, in the revised conclusion paragraph, the term “demonstrated” was removed, and main limitations of aRBCX approach were reported (lines 383-389).
We thank you for having given us the opportunity of revising the paper according to the referee’s comments.
Hoping that the revised version of our work might be suitable for publication in Journal of Clinical Medicine, I send my best regards.
Sincerely yours,
Caterina Giovanna Valentini
Transfusion Medicine Department
Fondazione Policlinico Universitario A. Gemelli IRCCS
I-00168 Roma, Italy
39-06-30154180
caterinagiovanna.valentini@policlinicogemelli.it
Round 2
Reviewer 1 Report
Comments and Suggestions for Authors
I do not have any more suggestions